# Tropism of the AAV6.2 Vector in the Murine Retina

**DOI:** 10.3390/ijms26041580

**Published:** 2025-02-13

**Authors:** Ryo Suzuki, Yusaku Katada, Momo Fujii, Naho Serizawa, Kazuno Negishi, Toshihide Kurihara

**Affiliations:** 1Laboratory of Photobiology, Keio University School of Medicine, Shinjuku-ku, Tokyo 160-0016, Japan; ryosuzuki0426@keio.jp (R.S.); yusakukatada@keio.jp (Y.K.); momof@keio.jp (M.F.); naho.serizawa@keio.jp (N.S.); 2Department of Ophthalmology, Keio University School of Medicine, Shinjuku-ku, Tokyo 160-8582, Japan; kazunonegishi@keio.jp

**Keywords:** gene therapy, RP, IRD, AAV, AAV6.2, Müller cells

## Abstract

Retinitis pigmentosa (RP) is a progressive inherited retinal dystrophy (IRD) that primarily affects rod photoreceptor cells, leading to the degeneration of photoreceptors and the gradual loss of vision. While RP is one of the most studied IRDs, other neurodegenerative diseases affecting the retina and optic nerve, such as glaucoma, also involve common mechanisms of cellular stress and degeneration. Current therapeutic approaches under investigation include gene therapy, retina prosthesis, and neuroprotection. Among these approaches, gene therapy has shown promise, though challenges related to viral vector tropism and transduction efficiency persist. The adeno-associated virus (AAV) vector is commonly employed for gene delivery, but novel serotypes and engineered variants are being explored to improve specificity and efficacy. This study evaluates the gene transfer efficiency of the AAV6.2 vector following intravitreal injection into the murine retina. Male C57BL/6 mice (9 weeks old) were intravitreally injected with 1 µL of AAV2-CMV-EGFP, AAV6-CMV-EGFP, or AAV6.2-CMV-EGFP at a titer of 3.2 × 10^12^ vg/mL per eye. Retinal transduction was assessed using in vivo fluorescence imaging, flat-mount imaging, and immunohistochemistry. EGFP expression in retinal ganglion cells, Müller cells, amacrine cells, and bipolar cells was quantitatively analyzed. All three AAV serotypes effectively transduced retinal ganglion cells, but AAV6.2 exhibited enhanced transduction in Müller cells and other neuronal retinal cells, including bipolar and amacrine cells. AAV6.2 demonstrated more localized expression around retinal blood vessels compared to the diffuse expression observed with AAV2. Immunohistochemical analysis revealed that AAV6.2 had significantly higher transduction efficiency in Müller cells (*p* < 0.001) compared to AAV2 and AAV6. AAV6.2 shows superior transduction efficiency in Müller cells, positioning it as a promising vector for gene therapies targeting retinal degenerative diseases such as RP. Its ability to effectively transduce Müller cells suggests potential applications in neuroprotection and gene replacement therapies.

## 1. Introduction

Inherited retinal dystrophies (IRDs) represent a clinically and genetically heterogeneous group of progressive retinal diseases characterized by the dysfunction and eventual loss of photoreceptors and/or retinal pigment epithelium (RPE). These conditions lead to visual impairment, including loss of color and night vision, peripheral visual defects, and, in severe cases, blindness. Among the IRDs, retinitis pigmentosa (RP) is the most prevalent and well-studied disorder, affecting approximately 1 in 4000 individuals. RP is a progressive degenerative disease that primarily affects the rod photoreceptors responsible for night and peripheral vision, eventually leading to the involvement of cone photoreceptors, which results in the loss of central vision. The hallmark feature of RP is the gradual constriction of the visual field, which can culminate in total blindness in advanced stages [1].

While RP is one of the most studied IRDs, other neurodegenerative conditions, such as glaucoma, also involve common mechanisms of cellular stress and degeneration. Glaucoma is primarily a disease of the optic nerve, characterized by the loss of retinal ganglion cells; however, recent studies have highlighted the role of Müller cells and their response to neurodegenerative stress in the disease process. Given these shared pathways, targeting Müller cells represents a promising strategy for addressing retinal and optic nerve diseases caused by diverse stresses.

Recent advancements in gene therapy, retina prosthesis, and neuroprotection have shown promise for treating IRD. Gene therapy strategies can be categorized into gene-specific approaches, such as gene augmentation and gene editing using CRISPR/Cas9, and gene-agnostic approaches like optogenetics, which aim to restore visual function by introducing microbial rhodopsins into retinal cells [2]. Neuroprotective therapy delivers neuroprotective factors to the retina. Moreover, recent studies show the possibility that Müller cells can be reprogrammed to regenerated photoreceptors by targeting the expression of genes or the application of exogenous factors [3]. Optogenetic strategies typically target bipolar cells and retinal ganglion cells, while neuroprotective therapies often focus on Müller cells. Gene therapies are designed to target various retinal cell types, employing different AAVs tailored to each specific cell type.

Given the heterogeneity of RP, different adeno-associated virus (AAV) serotypes have been explored to optimize gene delivery to specific retinal cell types. AAVs are commonly used due to their low immunogenicity and ability to achieve long-term gene expression. Beyond RP, their potential application in other retinal and optic nerve diseases, such as glaucoma, warrants further investigation. Among them, AAV2, AAV7m8, AAV8, and AAVDJ are reported to be highly efficient in infecting retinal ganglion cells, and AAV6 is reported to have high infection efficiency in the inner nuclear layer [4,5,6]. Additionally, AAV2 and AAV6 are reported to have high infection efficiency [5].

There are many different serotypes of AAV, and there is a need for high-performance AAVs for intravitreal injection. Therefore, it is important to continuously evaluate new AAV vectors. In this study, we focused on AAV6.2, which was produced by the F129L point mutation of AAV6.

AAV6.2 has a single nucleotide substitution, where the 129th amino acid phenylalanine (F) is replaced with leucine (L). This alteration characterizes AAV6.2 with high infection efficiency, particularly in airway epithelial cells, and it has been utilized in pulmonary, hepatic, and inner ear studies. Although there are no existing reports of AAV6.2 infecting retinal tissues, it is derived from AAV6—a serotype known for its efficacy in the retina. Therefore, we hypothesized that AAV6.2 might exhibit enhanced retinal infection efficiency. To explore this, we administered AAV6.2 to the eyes of mice and conducted a comparative analysis of retinal tissue tropism with AAV2 and AAV6, thereby investigating AAV6.2’s potential applicability for retinal gene therapy [7,8].

## 2. Results

### 2.1. In Vivo Fundus Fluorescence Imaging

Following intravitreal injections of AAV2-CMV-EGFP, AAV6-CMV-EGFP, and AAV6.2-CMV-EGFP, fundus fluorescence imaging revealed significant EGFP expression in the retinas of live mice starting at one week post-injection. Fluorescent signals continued to increase, peaking at four weeks and remaining stable for at least six weeks for all AAV serotypes (Figure 1a). AAV2 produced uniform GFP expression across the retina, while AAV6 and AAV6.2 demonstrated more localized, patchy fluorescence patterns (Figure 1a). Quantitative analysis showed that no significant difference in overall fluorescence intensity was observed among the three vectors (Figure 1b).

### 2.2. Flat-Mount Imaging

In the previous experiment, we used in vivo imaging, but this time, we used flat-mount staining to take a closer look. Approximately 6 weeks after the intravitreal injection, EGFP expression was observed across the ganglion cell layer (GCL) in all serotypes (*n* = 3 retinas) (Figure 2). The retinas injected with AAV2 exhibited uniform expression of EGFP (Figure 2a). In contrast, with AAV6 and AAV6.2 injections, EGFP signaling was notably concentrated in specific regions, particularly surrounding the blood vessels (Figure 2b,c).

### 2.3. Histological Analysis of Retinal Cross-Sections

Histological analysis of retinal cross-sections showed EGFP fluorescence in ganglion cells, bipolar cells, amacrine cells, and Müller cells in all serotypes (Figure 3). The number of fluorescent cells was quantified within a 212.13-µm square region on each retinal cross-section and then averaged. No significant transduction differences were noted in ganglion cells (*p* = 0.51) (Figure 3A–D).

A co-labeling approach using Glutamine Synthetase (GS) identified that transduction with AAV6.2 showed significantly higher transduction efficiency in Müller cells compared to AAV2 and AAV6 (*p* < 0.0001) (Figure 3E–H). Moreover, a co-labeling approach using PKCα identified that transduction with AAV6.2 was significantly more efficient in bipolar cells compared to AAV2 (*p* < 0.01) (Figure 3I–L).

There are different types of amacrine cells in the murine retina, and Syntaxin specifically labels amacrine cells [9]. Co-labeling with Syntaxin in amacrine cells showed that AAV2 was significantly more efficient than AAV6 (*p* < 0.05) (Figure 3M–P). Additionally, co-labeling with Syntaxin in amacrine cells indicated that AAV6.2 demonstrated a trend toward higher efficiency compared to AAV6 (*p* = 0.12) (Figure 3M–P). Individual channel images are provided in Section A.1, Section A.2 and Section A.3 for clarity.

## 3. Discussion

This study aimed to evaluate the retinal tropism and transduction efficiency of the AAV6.2 vector in comparison with the well-characterized AAV2 and AAV6 vectors. Our findings demonstrated that AAV6.2, while derived from AAV6 through a single-nucleotide mutation (F129L), exhibited unique and advantageous tropism in the murine retina, particularly in Müller cells and other non-neuronal retinal cells. This suggests that AAV6.2 holds significant potential for gene therapy applications targeting retinal degenerative diseases, such as RP. The results for AAV2 were consistent with the results of previous studies [6] in that it mainly infected retinal ganglion cells.

Next, we turn to AAV6.2. The AAV6.2 examined in this study is the F129L point mutant of AAV6. AAV6 is a type of adeno-associated virus found in human cells, and AAV6 has been reported to be efficiently expressed in airway epithelial cells. Furthermore, AAV6 is less immunogenic than AAV2, which may be highly advantageous in the treatment of chronic diseases such as cystic fibrosis [10,11,12]. AAV6.2 is a point mutation of AAV6 that changes the phenylalanine at position 129 to leucine to increase its affinity for lung cells and enhance the efficiency of infection. AAV6 has been reported to strongly infect Müller cells in the rat retina.

AAV6.2 tended to be expressed faster than AAV2 and AAV6. The expression cassettes are identical, and amino acid mutations on the capsid surface affect the expression rate, which could be due to the effect of capsid mutations. The first contact of AAV with the retina is at the inner limiting membrane (ILM), the inner edge of Müller cells, suggesting that the high efficiency of AAV6.2 expression in Müller cells is related to its expression rate.

Moreover, compared to AAV2, AAV6.2 showed a patchy distribution of expression in whole mounts, and sections showed that expression was predominantly in Müller cells. AAV6.2 expression tended to follow blood vessels, suggesting a strong affinity between Müller cells and retinal vessels. The observed localized expression of EGFP along blood vessels in flat-mount retinas may reflect Müller glia protrusions, as reported previously [6]. This spatial specificity suggests potential interactions with vascular-associated cells, such as endothelial cells or pericytes, which could have significant implications for retinal vascular diseases. In particular, the expression along retinal vessels might make AAV6.2 suitable for gene therapy targeting Müller cells [3] or for addressing retinal vascular lesions, such as anti-angiogenic therapy for diabetic retinopathy or therapeutic interventions in retinal vein occlusion [13].

One of the most well-studied AAV2 receptors is heparan sulfate proteoglycan [14], and fibroblast growth factor receptor 1 (FGFR1) [15] and αVβ5 integrin [16] have also been shown to act as co-receptors, promoting viral binding and uptake. Heparan sulfate proteoglycans (HSPGs) are a group of molecules in which heparan sulfate (HS) is covalently bound to a core protein, and they are present on the surface of the cell membrane and in the extracellular matrix of almost all animal cells [17]. In other words, AAV2 has the ability to bind to HS. It has also been found that AAV6.2 has the ability to bind to HS, and it has been suggested that AAV6.2 may also have the potential to bind to HSPG [18]. Since HSPG is present on the surface of the cell membrane and in the extracellular matrix of almost all animal cells, it was thought that AAV6.2 may have the same broad range of infection as AAV2.

Above all, AAV6.2 was more efficiently expressed in Müller cells than AAV2 and AAV6. AAV6.2 had a higher affinity for glial-supporting cells, such as cochlear-supporting cells. Improved heparin binding of AAV6.2 may have facilitated the passage of the inner boundary membrane barrier composed of heparan sulfate proteoglycans (HSPGs) [19].

The distinct distribution of AAV6.2 within the retina underscores its potential utility in treating IRDs beyond RP, including conditions that involve Müller cell dysfunction [20] or where inner retinal cells are key therapeutic targets. Given the increasing interest in Müller cells for gene therapy applications—such as their role in neuroprotective and reprogramming therapies [21]—AAV6.2’s efficacy in these cells positions it as a promising vector for further development. Its ability to transduce a broader range of retinal cells, including bipolar and amacrine cells, further expands its applicability in optogenetic approaches. Moreover, the high transduction efficiency observed in non-neuronal cells such as Müller cells suggests that AAV6.2 could be explored for gene replacement therapies in cases where Müller cells are directly affected, as seen in certain hereditary retinal dystrophies [22]. These findings suggest that AAV6.2 may serve as a versatile platform for various gene therapy strategies, from delivering neuroprotective agents to driving gene reprogramming efforts aimed at regenerating functional retinal tissue [23].

Despite the promising results, several limitations should be acknowledged. First, this study was conducted in murine models, and while the retinal structure of mice shares similarities with the human retina, there are important differences, particularly in the size and organization of retinal layers. Future studies should assess the tropism of AAV6.2 in larger animal models and, eventually, in human retinal tissue to validate its clinical applicability.

Additionally, while AAV6.2 showed a favorable transduction profile in this study, further research is needed to investigate its safety, particularly with respect to immunogenicity and long-term expression. Future studies will assess the immunogenic profile of AAV6.2 using both in vitro and in vivo models. Notably, previous research has reported that high intracellular concentrations of transfected GFP do not induce toxic effects [24]. While this finding supports the general safety of GFP expression, the potential for immune responses against AAV capsids remains a concern in gene therapy applications. Therefore, the immunogenic profile of AAV6.2 should be comprehensively characterized in preclinical and clinical settings to ensure its safety and efficacy for therapeutic use.

## 4. Materials and Methods

### 4.1. Animals

All animal experiments were conducted in accordance with protocols approved by the Institutional Animal Care and Use Committee of Keio University School of Medicine (#27-015-22). Male C57BL/6 mice (9 weeks old) were used for all experiments. The mice were maintained under a 12 h light–dark cycle, with food and water available ad libitum.

### 4.2. Vector Production and Purification

We used the ready-made AAV vectors, specifically AAV 2, 6, 6.2-CMV-EGFP-WPRE (Vector Builder, Chicago, IL, USA). Final preparations were dialyzed against phosphate-buffered saline (PBS) and stored at −80 °C.

### 4.3. Virus Injection

The mice were anesthetized using a mixture of medetomidine (0.3 mg/kg), midazolam (4 mg/kg), and butorphanol (5 mg/kg) (MMB) and placed on a heating pad that maintained their body temperature at 35–36 °C throughout the experiments. An aperture was made next to the limbus through the sclera with a 30-gauge disposable needle, and a 33-gauge unbeveled blunt-tip needle on a Hamilton syringe was introduced through the scleral opening into the vitreous space for intravitreal injections. Each eye received 1 µL of vehicle (PBS) or vector at a titer of 3.2 × 10^12^ vg/mL.

### 4.4. In Vivo Fundus Fluorescence Imaging

The mice were anesthetized with MMB mixture, and the pupils were dilated with a mixed solution of 0.5% tropicamide and 0.5% phenylephrine (Mydrin-P; Santen, Osaka, Japan). Live fundus images after the AAV injection were captured using a stereomicroscope (M165-FC; Leica Microsystems, Wetzlar, Germany) with a glass slide and water to cancel the refraction. Fluorescence intensity was measured from photographs taken under the same conditions using ImageJ2 software version 2.90/1.53t (National Institutes of Health, Bethesda, MD, USA).

### 4.5. Retinal Flat-Mount Preparation and Cryosections

Six weeks after vector injection, the mice were humanely euthanized. The eyes were removed and fixed with 4% paraformaldehyde in PBS for 1 h, and the cornea and lens were removed. To make flat mounts, the entire retina was carefully dissected from the eyecup, and radial cuts were made from the edges to the retinal equator. For cryosections, the eyecups were washed in PBS followed by immersion in 30% sucrose in the same buffer overnight. The eyecups were then cryoprotected in O.C.T. Compound (Sakura Finetek Japan, Tokyo, Japan) and cryosectioned into 12 µm thick transverse retinal sections.

### 4.6. Immunohistochemistry

Transduced mouse retinas were dissected and fixed in 4% paraformaldehyde for 30 min at room temperature. The retinas were incubated in PBS with 1% Triton X-100 and 0.5% Tween 20 for 1 h at room temperature, followed by incubation in 10% normal goat serum for 1 h at room temperature. The retinas were then incubated overnight at 4 °C with primary antibodies: RBPMS (1:500; Abcam, Cambridge, UK), PKCα (1:500; Abcam), Syntaxin (1:100; Abcam), glutamine synthetase (GS) (1:500; Merck, Boston, MA, USA), or GFP (1:500; Thermo Fisher Scientific, Waltham, MA, USA) in blocking buffer. Secondary antibodies conjugated with Alexa TM647, Alexa TM555, or Alexa TM488 (1:1000; Molecular Probes, Eugene, OR, USA) were applied for 1 h at room temperature. The retinas were then flat-mounted, and the sections were mounted on slide glass.

### 4.7. Fluorescent Imaging

Retinal flat-mount images were obtained using a fluorescence microscope (THUNDER Imager 3D Cell Culture, Leica Microsystems, Mumbai, India). Fluorescent images of the cryosections were obtained using a confocal microscope (LSM980, Carl Zeiss, Jena, Germany). In total, 3 retinas were used per vector, and 4 images were taken of each.

The number of fluorescent cells, which were labeled with DAPI, EGFP, and other antibodies, was quantified within a 212.13 µm square region on each retinal cross-section and then averaged.

### 4.8. Statistical Analysis

Statistical analysis was performed using Prism software version 9 for Windows (Dotmatics, Boston, MA, USA). The criterion for statistical significance was *p* < 0.05. An unpaired *t*-test and one-way ANOVA with Bonferroni correction were performed.

## 5. Conclusions

In conclusion, AAV6.2 demonstrates enhanced transduction efficiency in Müller cells, suggesting that it could play a pivotal role in advancing gene therapies for retinal degenerative diseases, including RP. Its ability to target non-neuronal cells opens up new possibilities for neuroprotective and regenerative therapies. Further studies in larger models and clinical trials will be critical to fully assess the therapeutic potential of AAV6.2 and its safety for human applications.

## Figures and Tables

**Figure 1 ijms-26-01580-f001:**
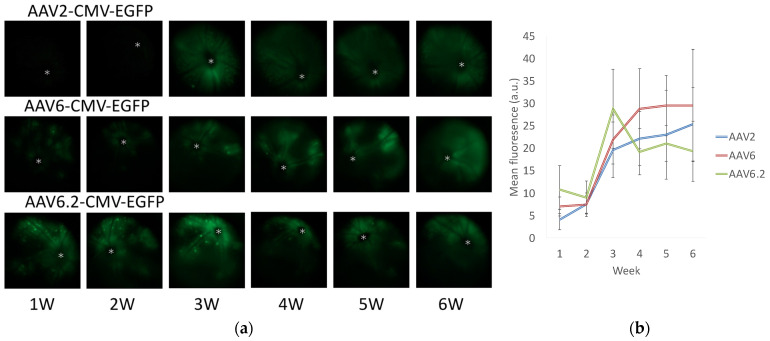
Time course of EGFP expression in the retina following AAV intravitreal injection. (**a**) Representative fundus photographs illustrating EGFP expression in the retinas of live mice at 1, 2, 3, 4, 5, and 6 weeks post-injection. The mice were injected intravitreally with AAV2-CMV-EGFP, AAV6-CMV-EGFP, or AAV6.2-CMV-EGFP. Asterisks indicate optic discs. (**b**) Quantification of EGFP fluorescence intensity from fundus fluorescent photographs. The mean fluorescence values were measured over the circular imaging range. Error bars represent the standard error of the mean. a.u.: arbitrary unit.

**Figure 2 ijms-26-01580-f002:**
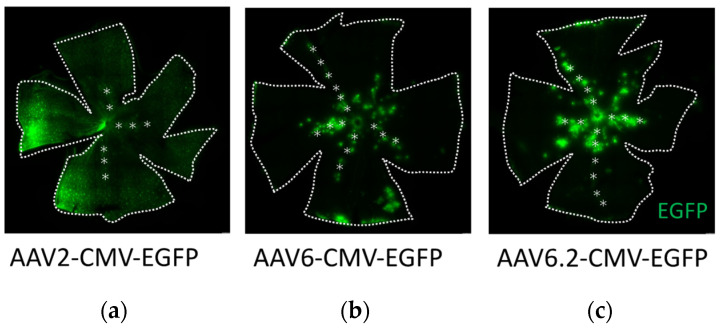
Distribution of EGFP expression. (**a**–**c**) Representative confocal images of retinal flat-mounts at 6 weeks post-injection, showing EGFP expression following intravitreal injection of AAV2-CMV-EGFP (**a**), AAV6-CMV-EGFP (**b**), and AAV6.2-CMV-EGFP (**c**). Asterisks indicate retinal vessels. Blood vessels were identified based on their characteristic appearance under bright-field microscopy.

**Figure 3 ijms-26-01580-f003:**
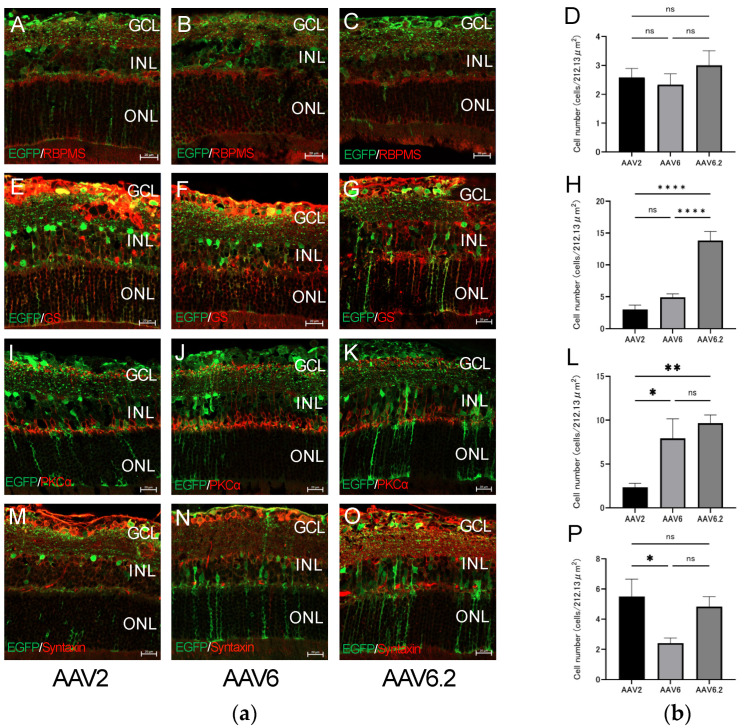
Immunostaining of each retinal cell type of AAV vectors intravitreally injected into the mouse retina and evaluation of EGFP expression. (**a**) Immunohistochemistry on transverse retinal cryosections from murine retinas with intravitreal injection of AAV2-CMV-EGFP, AAV6-CMV-EGFP, AAV6.2-CMV-EGFP for (**A**–**C**) retinal ganglion cells labeled with RBPMS, (**E**–**G**) Müller cells labeled with GS, (**I**–**K**) bipolar cells labeled with PKCα, and (**M**–**O**) amacrine cells labeled with Syntaxin. The individual channels and DAPI staining can be found in Appendix A. (**b**)The number of fluorescent cells was quantified within one field of view on each retinal cross-section and then averaged. (**D**) Quantification of the EGFP-positive rate in RBPMS-positive cells. (**H**) Quantification of the EGFP-positive rate in GS-positive cells. (**L**) Quantification of the EGFP-positive rate in PKCα-positive cells. (**P**) Quantification of the EGFP-positive rate in Syntaxin-positive cells. The criterion for statistical significance was *p* < 0.05, and an unpaired *t*-test and one-way ANOVA with Bonferroni correction was performed. *n* = 12 images; *n* = 3 retinas. Of note, 4 images were taken per retina. Error bars represent the standard error of the mean. ns, not significant. * *p* < 0.05. ** *p* < 0.01. **** *p* < 0.0001. GCL, ganglion cell layer; INL, inner nuclear layer; ONL, outer nuclear layer. Scale bars, 20 µm in (**A**–**C**,**E**–**G**,**I**–**K**,**M**–**O**).

## Data Availability

The data presented in this study are available from the corresponding author upon request. The data are not publicly available due to intellectual property restrictions.

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
