# Peer review of "Tropism of the AAV6.2 Vector in the Murine Retina"

_ijms, 2025, doi:10.3390/ijms26041580_

Round 1
Reviewer 1 Report
Comments and Suggestions for Authors
The present paper provides evidence of the potentiality of the new viral vector AAV-6.2 to transduce in Muller cells and other retinal neurons like bipolar and amacrine cells. This result seems to suggest the advantage to use this vector to deliver neuroprotective agents and increase the probability to cope with neurodegenerative diseases. The experimental protocol is clearly described and results are well presented. The main criticism is on the introduction where the authors describe RP , they have to clarify that the major mutations are in rod photoreceptors and the target of gene therapy have to be photoreceptors. Apparently the present research aims is targeting downstream events modulated by Muller cell in response to degenerative events. I think that they have to say from the beginning that the target is retinal neurodegenerative diseases due to a variety of stresses (i.e. glaucoma and more). I think that this point needs to be clarified from the beginning. It has been pointed out in the discussion. But the introduction is misleading.
Reviewer 2 Report
Comments and Suggestions for Authors
The manuscript “Tropism of AAV-6.2 vector in the murine retina” addresses an important problem of gene therapy. However, the following questions need to be addressed before the manuscript can be accepted:
Major concerns:
1. In the figure 2 blood vessels are indicated by asterisks. How was this determined? It would be more convincing if double immunostaining is used do demonstrate EGFP expression relative to blood vessels. It will be also interesting to address this expression pattern more in depth in the discussion – for example, what could this mean in the scope of translational significance.
2. In the Figure 3:
o it is hard to see co-localization of the green and purple colors. I would recommend preudocoloring purple with red so that co-localization between green and red is more obvious.
o Is there a reason why the green channel on images E vs I and G vs K is identical? Was it a triple immunostaining? It might be good to demonstrate not overlayed images along with the overlayed ones.
o it is not clear what was measured in the bar graphs. The legend states that it is the number of fluorescent cells. Did you mean double fluorescent cells? How was the counting performed without nuclear staining? It is pretty hard to distinguish cells that are close to each other and overexposed.
3. The first paragraph of the discussion should be deleted, as this is clearly from the paper template
4. While the authors address the lack of immunogenicity studies in the discussion, the original manuscript would benefit from these data.
Minor concerns:
- Page 2, line 47. Typo in the word delivers
- Page 2, line 77. I would change “up to six weeks” to “at least six weeks”
- Figure 1 – the legend doesn’t state, what asterisks denote.
- Page 5, line 145. The sentence is lacking something (probably should state “has been reported to be efficiently expressed in airway epithelial cells”)
- Methods, immunohistochemistry: when listing antibodies, change “and” to “or”. Also, should it be Alexa 647 instead of 648?
Round 2
Reviewer 1 Report
Comments and Suggestions for Authors
The authors changed the test according to the suggestion in an appropriate way.
Author Response
We sincerely appreciate your feedback. We are glad to hear that our revision appropriately addressed your suggestion. Thank you for your valuable input in improving our manuscript.
Reviewer 2 Report
Comments and Suggestions for Authors
Thank you for the revision. The manuscript has been significantly improved and just a few minor points remain:
1. Typo in the word recent on line 59; the word factor should be plural on line 61
2. Please, shorten the edit of Figure 2 legend. It is sufficient just to state "Blood vessels were identified based on their characteristic appearance under bright-field microscopy [6,9].
3. Fig. 3 legend (Line 150) - probably add that the individual channels and DAPI staining can be found in Appendix
4. I would not separate "the way of counting fluorescent cells (lines 308-311) as a separate section. I would combine it with the section "fluorescent imaging"
